# Defect-Rich Heterogeneous MoS_2_/rGO/NiS Nanocomposite for Efficient pH-Universal Hydrogen Evolution

**DOI:** 10.3390/nano11030662

**Published:** 2021-03-08

**Authors:** Guangsheng Liu, Kunyapat Thummavichai, Xuefeng Lv, Wenting Chen, Tingjun Lin, Shipeng Tan, Minli Zeng, Yu Chen, Nannan Wang, Yanqiu Zhu

**Affiliations:** Guangxi Institute Fullerene Technology (GIFT), Guangxi Key Laboratory of Processing for Non-Ferrous Metals and Featured Materials, School of Resources, Environment and Materials, Guangxi University, Nanning 530004, China; Jesson_lgs@163.com (G.L.); kt302@exeter.ac.uk (K.T.); Betsy_Lv@163.com (X.L.); wt15578102973@126.com (W.C.); w15260260610@163.com (T.L.); 1739200127@st.gxu.edu.cn (S.T.); MinliZeng_97@163.com (M.Z.); yc465@exeter.ac.uk (Y.C.); Y.zhu@exeter.ac.uk (Y.Z.)

**Keywords:** electrocatalyst, hydrogen evolution reaction, molybdenum disulfide, reduced graphene oxide, nickel sulfide

## Abstract

Molybdenum disulfide (MoS_2_) has been universally demonstrated to be an effective electrocatalytic catalyst for hydrogen evolution reaction (HER). However, the low conductivity, few active sites and poor stability of MoS_2_-based electrocatalysts hinder its hydrogen evolution performance in a wide pH range. The introduction of other metal phases and carbon materials can create rich interfaces and defects to enhance the activity and stability of the catalyst. Herein, a new defect-rich heterogeneous ternary nanocomposite consisted of MoS_2_, NiS and reduced graphene oxide (rGO) are synthesized using ultrathin αNi(OH)_2_ nanowires as the nickel source. The MoS_2_/rGO/NiS-5 of optimal formulation in 0.5 M H_2_SO_4_, 1.0 M KOH and 1.0 M PBS only requires 152, 169 and 209 mV of overpotential to achieve a current density of 10 mA cm^−2^ (denoted as η_10_), respectively. The excellent HER performance of the MoS_2_/rGO/NiS-5 electrocatalyst can be ascribed to the synergistic effect of abundant heterogeneous interfaces in MoS_2_/rGO/NiS, expanded interlayer spacings, and the addition of high conductivity graphene oxide. The method reported here can provide a new idea for catalyst with Ni-Mo heterojunction, pH-universal and inexpensive hydrogen evolution reaction electrocatalyst.

## 1. Introduction

Hydrogen energy is expected to be an ideal energy source in the future due to its high energy density and zero pollution [1,2]. Electrolytic water is a simple and effective method to produce high purity hydrogen. High efficiency of hydrogen evolution reaction can be achieved by efficient electrocatalysts such as Pt, which can decrease the activated barrier and accelerate the reaction rate. However, noble electrocatalysts cannot be widely used in industrial hydrogen production because of their high price and scarcity. In view of the above, the development of high efficiency non-noble metal electrocatalysts is of great significance to boost the development of hydrogen energy resources.

Two-dimensional transition metal sulfides (TMDs), such as MoS_2_, WS_2_, NiS_2_ and CoS_2_, have been widely investigated due to their excellent electrocatalytic properties [3,4,5,6]. Among the materials, MoS_2_ shows superior catalytic activity and is considered as a promising catalyst due to its low adsorption energy between unsaturated sulfur atoms and hydrogen atoms [7]. Unfortunately, few active sites, intrinsic low conductivity and insufficient stability hinder the HER activity of MoS_2_ [8]. To solve these problems, various effective methods such as defect and interfaces engineering, structural design and elements doping technique have been introduced for improving electrocatalytic activity of MoS_2_. Considering that the active site was derived from the Mo-S of edge of MoS_2_ catalysis, surface defects or interfaces are essential to improve the electrocatalytic activity of catalysts [9,10,11]. Various defect-rich MoS_2_ electrocatalysts has been previously reported, including doping heteroatom to form various M-Mo-S (M = Co, Ni) structure [12,13,14], interlayer-expanded MoS_2_ nanosheets [15], N-doped MoS_2_ nanosheet [16] and addition of excessive sulfur sources [17]. These modified MoS_2_ structures with poor crystallinity are easily dissolved in acidic solution, resulting in poor electrochemical stability. In addition, due to slow HER kinetics and poor stability in alkaline medium, it is still remaining a big challenge to produce MoS_2_-based catalysts to meet the industrial requirements.

The kinetics of HER in acidic solution is 2–3 times faster than in alkaline solution due to the large adsorption energy of OH^−^ on the surface of catalyst in alkaline medium and high dissociation energy barrier between MoS_2_ and water [18,19]. Modifying the MoS_2_ structure helps to reduce its binding energy to hydroxyl, resulting in the improvement of catalyst HER performance in alkaline medium. Previous studies have demonstrated that NiS_x_ has excellent stability under alkaline conditions, and can form a heterostructure with MoS_2_ nanosheets to achieve both high catalytic performance and play a protective role for MoS_2_ [20].

Apart from HER performance, the charge conductivity of MoS_2_-based catalyst is also another key performance for electrocatalytic materials. Nickel and copper foam materials have been used as conductive substrates and supporting materials, then in situ growth MoS_2_ nanosheets on their surfaces [21,22,23,24]. This strategy has shown excellent performance in HER and OER processes. Unfortunately, the metal foam in these self-supporting catalysts could be dissolved by acid, which severely limits their widespread use. Carbon materials have excellent electrical conductivity and stability in a wide pH range, such as reduced graphene oxide (rGO), carbon fiber paper and carbon cloth. The reported CoS_2_@MoS_2_/rGO [19], MoS_2_/rGO [25] and Mn/MoS_2_/rGO [26] catalysts have good catalytic stability and electrocatalytic activity, which owes to the synergetic effect of rGO intrinsic high charge conduction characteristics [23]. Therefore, it is an effective strategy to grow MoS_2_ on rGO using rGO as conductive substrate.

Herein, defect-rich multiphase MoS_2_/rGO/NiS nanocomposite catalyst was successfully synthesized using αNi(OH)_2_ nanowires with excellent catalytic performance and stability over a wide range of pH. The existence of abundant defects and heterogeneous interfaces was confirmed by X-ray diffraction analysis (XRD), Raman spectrum and high resolution transmission electron microscopy (HRTEM). Furthermore, the optimal MoS_2_/rGO/NiS-5 nanocomposite exhibited very low overpotential and good stability in 0.5 M H_2_SO_4_ (η_10_ = 152 mV), 1.0 M KOH (η_10_ = 169 mV) and 1.0 M PBS (η_10_ = 209 mV), respectively.

## 2. Results and Discussion

The MoS_2_/rGO/NiS nanocomposite with rich defects and heterogeneous interfaces was synthesized by two-step hydrothermal method, which is shown in Figure 1. (The detailed synthesized steps were in supporting information). L-cysteine hydrochloride (L-ch, HSCH_2_CHNH_2_COOH·HCl·H_2_O) can conjugate well with graphene oxide (GO) due to its multifunctional groups (-SH, -NH_2_, -COO). When heated, L-ch can not only be acted as a reductant to reduce Mo^6+^ to Mo^4+^, but also expand the interlayer spacing of MoS_2_ by releasing NH^4+^ and in situ intercalating into the interlayer of MoS_2_. The αNi(OH)_2_ adsorbed on the GO was sandwiched between the GO and the MoS_2_ nanosheets and was vulcanized into NiS, while the GO was also in situ reduced to reduced graphene oxide (rGO) during the hydrothermal process. Furthermore, the weakly acidic conditions formed by the dissolution of L-ch can dissolve a small amount of αNi(OH)_2_ and reduce the dissolved Ni^2+^ to Ni^0^ and then dope it into MoS_2_.

The crystal phases change of MoS_2_ based samples before and after doping GO and αNi(OH)_2_ are shown in Figure 2a. The diffraction peaks at 33.76°, 35.12° and 57.76°, correspond to (101), (102) and (110) plane of 2H-MoS_2_ (JPCDS no. 37-1492), respectively. These peaks are broad, indicative of the poorly crystallization of the samples. The standard (002) peak of pristine 2H-MoS_2_ at 14.4° are not observed in our as-prepared MoS_2_ samples. Instead, a peak at 9.41° is obtained, which can be ascribed to the insertion of NH^4+^ into MoS_2_ structure, suggesting an enlarged d value [27]. According to Bragg’s Law (nλ = 2dsinθ), the spacing between two adjacent S-Mo-S layers are calculated to be about 0.94 nm [28,29]. The (002) diffraction peak of the MoS_2_/rGO/NiS-5 shifts toward the lower 2-theta position at 8.5° compare with as-prepared MoS_2_. The d value is also calculated to be 1.06 nm, in accordance well with the results of TEM (Figure 3e). The reason of (002) peak shift is that graphene oxide may inhibit the growth of MoS_2_ crystal in the mixed catalyst. The diffraction pattern of MoS_2_/rGO/NiS-5 catalyst also matched well with NiS standard pattern (JCPDS no. 02-1280), suggesting αNi(OH)_2_ was transformed into NiS. Four broad peaks of NiS located at 30.37°, 34.65°, 45.70° and 53.93° were attributed to (100), (101), (102) and (110) planes, respectively. The XRD spectrum of αNi(OH)_2_ nanowires is shown in Appendix A.

Furthermore, the structural characteristics of the MoS_2_/rGO/NiS-5 can be obtained from the Raman spectrum in Figure 2. The main peaks at ~402, ~371 and ~296 cm^−1^ are attributed to A^1^_g_, E^1^_2g_ and E^1^_g_ mode of 2H MoS_2_, respectively, whereas the peak located at ~345 cm^−1^ belong to the modes of Ni-S [30,31]. More importantly, three sharp peaks in the range of 800–1000 cm^−1^ originate from the molecular structure of Mo_3_S_13_ located at edge of MoS_2_. It can be proven that there are abundant unsaturated Mo-S edge sites in MoS_2_/rGO/NiS-5 [19,30]. There are two characteristic peaks at ~1437 and ~1580 cm^−1^, which are G and D-band of GO, respectively. The degree of graphitization can be expressed by the strength ratio of D and G bands. As shown in Figure 2b, the I_D_/I_G_ of MoS_2_/rGO/NiS-5 (value of 1.11) is higher than that of GO (value of 0.83), indicating that the defects of GO increase with the reduction of oxygen-containing functional groups and GO is transformed into rGO [4,32].

Large specific surface area is a critical factor to determine the performance of the electrocatalyst of the electrode materials for the HER. The N_2_ isothermal adsorption desorption curve of the MoS_2_/rGO/NiS-5 shows type-IV isotherm (Appendix A), suggesting the existence of both micro-pores and meso-pores [33]. This specific surface area was 20.7 m^2^ g^−1^, which benefits from the abundant MoS_2_ nanosheets wrinkles and small particle size. The morphology of the MoS_2_/rGO/NiS-5 was shown in Figure 3 and Appendix A. A large number of as-prepared αNi(OH)_2_ nanowires are anchored uniformly on the GO substrate (Appendix A). It is observed that the MoS_2_/rGO/NiS-5 has a 3D crimp structure, which is attributed to the GO curling during hydrothermal process [34]. A large number of ultra-thin MoS_2_ nanosheets suspended on rGO are presented in Figure 3a–c and Appendix A. Such microstructure is conducive to exposing more marginal catalytic active sites. Further, the TEM images in Appendix A prove that the microstructure of the nanowires is maintained after the αNi(OH)_2_ nanowires are vulcanized to NiS. Moreover, the hybrid catalyst has a three-dimensional structure consisting of three layers of rGO, NiS and MoS_2_. The HRTEM image exhibited lattice fringe with spacing of 1.06 and 0.95 nm, which belonged to the (002) plane of MoS_2_ (Figure 3e,i). The lattice fringe of ~0.25 nm is indexed to the (101) plane of NiS (Figure 3f,h). Heterogeneous interface between MoS_2_ and NiS are observed (Figure 3d,g), resulting in rich-defects and disordered structure of our samples. Furthermore, the element mapping of the MoS_2_/rGO/rGO/NiS-5 hybrid sample is also carried out, which proves that Mo, Ni and S elements are uniformly distributed on rGO nanosheets (Figure 3j). The elemental composition ratio and spectral pattern of the hybrid catalyst are presented in Appendix A. 

The chemical states of elements in MoS_2_/rGO/NiS-5 nanocomposite were investigated by XPS. The survey XPS spectrum shows that the samples contain Mo, S, Ni, C, O, and N elements in Appendix A, which is verified by the results of EDS. From the percentage of atomic orbital, it can be known that 27.39% S 2p bonding with 13.85% Mo 3d and 3.36% Ni 2p orbitals (Appendix A), which indicates the existence of Mo-S defects. Comparing C 1s spectrum in GO sheets and MoS_2_/rGO/NiS-5 nanocomposite, the two peaks in GO at 286.7 eV and 287.5 eV are ascribed to graphitic sp^2^ carbon atom and C=O (Appendix A) [4,35,36]. However, these two peaks disappeared after hydrothermal treatment, which suggests that some of the oxygen-containing functional groups were reduced (Figure 4a). A new peak of C=N in C 1s spectrum of MoS_2_/rGO/NiS-5 appeared at 285.8 eV, which could be due to the part of NH^4+^ released by L-ch doping for rGO [37,38]. Two main Mo 3d peaks at 228.9 eV and 232.1 eV correspond to 3d_5/2_ and 3d_3/2_ of Mo^4+^, respectively, affirming the dominance of Mo^4+^ in MoS_2_/rGO/NiS-5 nanocomposite (Figure 4b). Another doublet peak is detected at 233.0 and 235.63 eV, which ascribe to the oxidation state of Mo^4+^. The peak of S 2s at 226.2 eV indicates the formation of Mo-S and Ni-S bonds [33,39]. For S 2p spectrum (Figure 4c), two types of distinct doublets (2P_3/2_, 2P_1/2_) were examined: (1) the peaks at 161.8 eV and 163.4 eV are divided and fitted to S 2p_3/2_ and S 2p_1/2_, respectively. (2) Another doublet at 162.9 and 164.5 eV are attributed to the bridging S_2_^2−^ ligands and the apical S^2−^ ligand of the [Mo_3_S_13_]^2−^, in agreement with the result of Raman spectrum [26,37,40]. Two resolved doublets (with a spin-orbits splitting of ~17.3 eV between 2p_3/2_ and 2p_1/2_) and two satellites are observed in the spectrum of Ni 2p (Figure 4d). The peaks located at 853.6 eV and 870.9 eV correspond to metallic Ni^0^ species in the sample; hence, the existence of Ni^0^ could be due to the reducibility of L-ch. The other two peaks at 856.0 and 873.7 eV in the spectrum are attributed to the presence of Ni^2+^ component [41,42].

The HER activity of the hybrid catalysts was investigated in different pH electrolyte solutions including 0.5 M H_2_SO_4_, 1.0 M KOH and 1.0 M PBS. Commercial 20% Pt/C was also examined in different electrolyte solutions. In acidic medium, the electrode coated with pristine MoS_2_ showed the worst catalytic activity, as shown in Figure 5a. Both MoS_2_/NiS-5 and MoS_2_/rGO/NiS-0 samples were exhibited higher catalytic activity and lower overpotential than the pristine MoS_2_. Hence, the trend of catalytic currents for these doped catalysts is MoS_2_/rGO/NiS-0 > MoS_2_/NiS-5 > MoS_2_. After adding different amounts of αNi(OH)_2_ into the MoS_2_/rGO, the activity increased with the increase of αNi(OH)_2_, whereas the activity decreased when the addition amount of αNi(OH)_2_ reached 7 mL. The optimal MoS_2_/rGO/NiS-5 exhibits excellent catalytic performance (η_10_ = 152 mV) and requires only 212 mV overpotential to achieve a current density of 100 mA cm^−2^ (denoted as η_100_) in acidic medium. Interestingly, when the current density is smaller than 40 mA cm^−1^, the polarization curves of MoS_2_/rGO/NiS-0 and MoS_2_/rGO/NiS-3 almost coincide. However, when the current density is more than 40 mA cm^−2^, the overpotential of MoS_2_/rGO/NiS-0 is larger than that of MoS_2_/rGO/NiS-3, which is probably because formed NiS improves the stability of MoS_2_. The overpotential of the MoS_2_/rGO/NiS-7 (η_10_ = 241 mV) is larger than other doped catalysts, which may be due to the fact that the large amount of NiS covers the active site of MoS_2_.

The MoS_2_/rGO/NiS-5 exhibits the best HER catalyst activity under alkaline condition, and the overpotential are 169 and 301 mV at current density of 10 and 100 mA cm^−2^, respectively (Figure 5d). Different from acid medium, the overpotential MoS_2_/rGO/NiS-7 exhibits the second lowest overpotential in alkaline media (η_10_ = 357 mV), which is related to the heterogeneous interface between NiS and MoS_2_ that facilitates the dissociation of water molecules. In neutral condition, the overpotential of MoS_2_/rGO/NiS-5 showed modest overpotential at 10 mA cm^−2^ (η_10_ = 209 mV) (Figure 5g).

The Tafel slope was fitted from the LSV curve to evaluate the HER kinetics of as-prepared samples. In 0.5 M H_2_SO_4_ (Figure 5b), MoS_2_/rGO/NiS-5 showed a very small Tafel slope of 54.3 mV/dec, which is close to 32.2 mV/dec of Pt/C. The small Tafel slope indicating that the HER first occur a very fast Volmer step (H_3_O^+^ + e^−^
→ H_ads_ + H_2_O), followed by a slow electrochemical desorption Heyrovsky step (H_ads_ + H_3_O^+^ + e^−^
→ H_2_ + H_2_O), and the desorption reaction of hydrogen was the rate limiting step of HER [38,43,44]. Moreover, the Tafel slopes for MoS_2_/rGO/NiS-X (X = 0, 3, 5, 7) and Pt/C were 190.4, 186.4, 91.6, 150.1 and 58.3 mV/dec in 1.0 M KOH solution, respectively (Figure 5e). The lower Tafel slope of MoS_2_/rGO/NiS-5 indicates the Volmer-Heyrovsky recombination mechanism of HER and the Volmer step of water splitting into protons and hydroxyls is the main step determining the reaction rate [45]. In 1.0 M PBS, the Tafel slopes of MoS_2_/rGO/NiS-X and Pt/C were 354.8, 215.7, 141.9, 216.3 and 69.7 mV/dec, respectively (Figure 5h). The results revealed that the rate limiting step of HER for as-prepared samples is slow Volmer step in a neutral medium [44].

The MoS_2_/rGO/NiS-5 nanocomposite suggests excellent stability in acidic, alkaline and neutral medium, as the polarization curve after 1000 cycles (Figure 5c,f,i), in which the overpotential has only increased by 17, 27 and 23 mV, respectively. Furthermore, chronopotentiometry of catalyst in different media were performed under 10 mA cm^−2^ current density of for 10 h (as shown in the insets image of Figure 5c,f,i). After ten hours, the overpotential increased by 14.4% (from 163 to 186 mV), 20.8% (from 226 to 273 mV) and 37.9% (from 208 to 287 mV) in 0.5 M H_2_SO_4_, 1.0 M KOH and 1.0 M PBS, respectively. These results indicate that the as-prepared MoS_2_/rGO/NiS-5 has a good stability, and it is more stable under acidic conditions than in alkaline and neutral conditions. Compared with other MoS_2_-based electrocatalysts reported recently, the MoS_2_/rGO/NiS-5 has superior performance in acidic and alkaline media (Table 1).

The electrochemical double-layer capacitance (C_dl_) of as-synthesized catalysts was measured to evaluate the electrochemical active surface areas (ECSA). Cyclic voltammograms (CVs) with various scan rates from 40–200 mV s^−1^ (Appendix A) were recorded to calculate C_dl_. The capacitance current (Δj = j_a_ − j_c_) and the scanning rates are fitted into a straight line, and the slope of the line is twice the value of C_dl_. The slope of MoS_2_/rGO/NiS-X (X = 0, 3, 5, 7) are 7.1, 9.01, 11.9 and 4.58 mF cm^−2^ in 0.5 M H_2_SO_4_, respectively (Figure 6a). Notably, MoS_2_/rGO/NiS-5 with the highest C_dl_ (5.95 mF cm^−2^) has the largest ECSA compared to other samples, which further demonstrates its excellent HER performance in acidic medium. The MoS_2_/rGO/NiS-5 (C_dl_ = 2.76 mF cm^−2^) exhibits better HER performance than MoS_2_/rGO/NiS-7 (C_dl_ = 2.46 mF cm^−2^) despite that they have similar ECSA in the alkaline medium (Figure 6c), indicating that the catalytic activity is determined by the coordination of hydrogen adsorption and the active sites of water adsorption/dissociation [46]. Similarly, MoS_2_/rGO/NiS-5 still has the largest C_dl_ value (1.66 mF cm^−2^) in neutral media (Figure 6e).

Electrochemical impedance spectra (EIS) was measured to further investigate the charge-transfer kinetics between electrocatalysts and electrolyte interface (Figure 6b,d,f). The EIS data are fitted to acquire the equivalent circuit patterns (insets in Figure 6b,d,f), where R_s_ is the solution resistance, the R_c_ at the high frequency region is related to the pores on the material surface and does not change with the change of overpotential, and the R_ct_ at the low frequency region represents the charge transfer impedance [43]. The results of the fitting data are shown in Appendix A. Evidently, MoS_2_/rGO/NiS-5 shows the lowest charge transfer resistance value of 19.75, 14.3 and 105.1 ohm in acidic, alkaline and neutral electrolytes, which demonstrate it’s the high electrochemical conductivity. Simultaneously, it can be clearly seen that the R_ct_ of MoS_2_/rGO/NiS-X in acid solution is much smaller than that of MoS_2_/NiS-5 (515.2 ohm), which is because the addition of rGO enhances the rate of electrons transfer between the electrode and catalytic active site. Furthermore, experiments show that 5 mL of αNi(OH)_2_ is the best addition to reduce the impedance of the catalyst. Importantly, the results of EIS studies are in agreement with above results of electrochemical performances studies. These results also confirmed that the abundant heterogeneous interface between NiS and MoS_2_ allow not only to expose dense catalytic sites but also improves the electron transfer ability of catalytic materials, which further enhance the catalytic activity of electrocatalyst. 

## 3. Conclusions

In this paper, defect-rich heterogeneous MoS_2_/rGO/NiS nanocomposite for HER has been successfully synthesized using αNi(OH)_2_ nanowires with excellent catalytic performance and stability over a wide range of pH. The effect of different amounts of NiS on the HER activity of MoS_2_/rGO were investigated. The results confirmed that defect-rich heterogeneous MoS_2_/rGO/NiS can be used as an efficient and stable HER catalyst under a universal range of pH electrolyte condition. The optimal MoS_2_/rGO/NiS-5 electrocatalysts exhibited good stability as well as very low overpotential in 0.5 M H_2_SO_4_ (η_10_ = 152 mV), 1.0 M KOH (η_10_ = 169 mV) and 1.0 M PBS (η_10_ = 209 mV), respectively. The outstanding HER performance of MoS_2_/rGO/NiS-5 catalyst can be ascribed mainly to the following three points: (1) the abundant heterogeneous interface between NiS and MoS_2_ can expose dense catalytic sites and accelerate electron transfer, which synergistically promotes electrocatalytic activity in a wide pH range; (2) the enlarged interlayer spacing of MoS_2_ exposes more marginal active edge sites; (3) the high intrinsic conductivity of rGO facilitates the rate of charges transfer between electrode and active catalytic site. This work can provide a new idea for designing Ni-Mo heterojunction materials as an efficient and low-cost pH-universal HER electrocatalyst.

## Figures and Tables

**Figure 1 nanomaterials-11-00662-f001:**
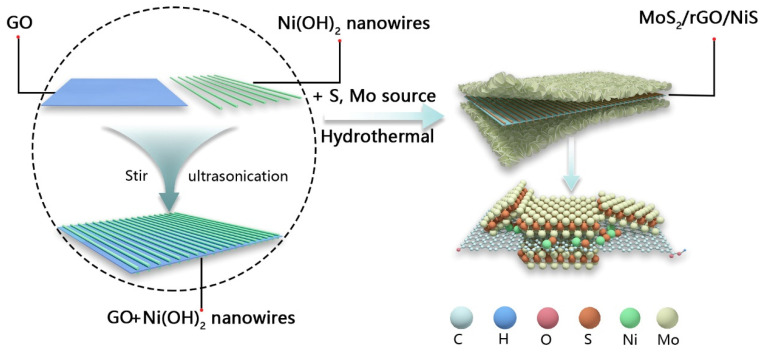
Schematic of synthesized process and structural of MoS_2_/rGO/NiS-X (where X is the volume of αNi(OH)_2_ added, X = 0, 3, 5, 7).

**Figure 2 nanomaterials-11-00662-f002:**
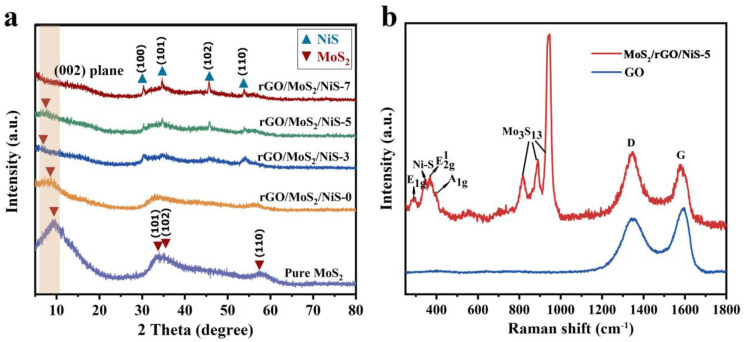
(**a**) XRD patterns of MoS_2_/rGO/NiS-X (X = 0, 3, 5, 7) nanocomposite and pure MoS_2_ (**b**) Raman spectrum of MoS_2_/rGO/NiS-5 nanocomposite and GO.

**Figure 3 nanomaterials-11-00662-f003:**
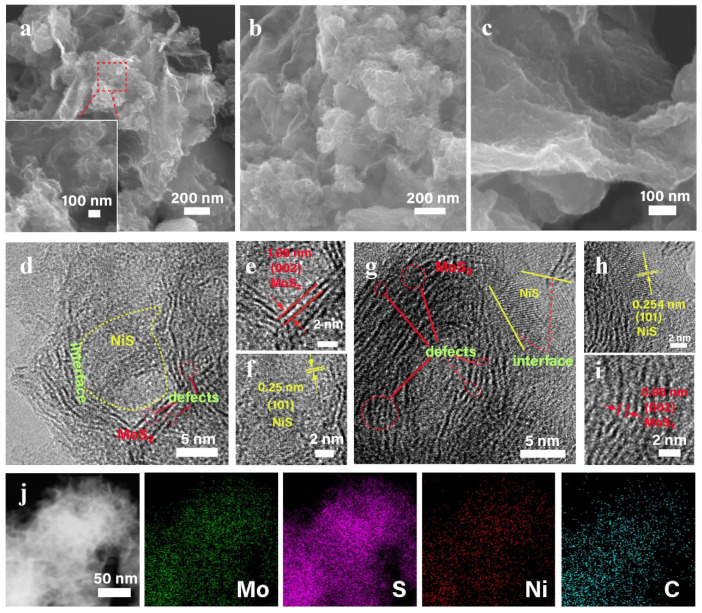
(**a**–**c**) SEM images of MoS_2_/rGO/NiS-5 nanocomposite. (**d**–**i**) HRTEM images of MoS_2_/rGO/NiS-5 nanocomposite. (**j**) HAADF-STEM and corresponding element mapping images in MoS_2_/rGO/NiS-5 nanocomposite.

**Figure 4 nanomaterials-11-00662-f004:**
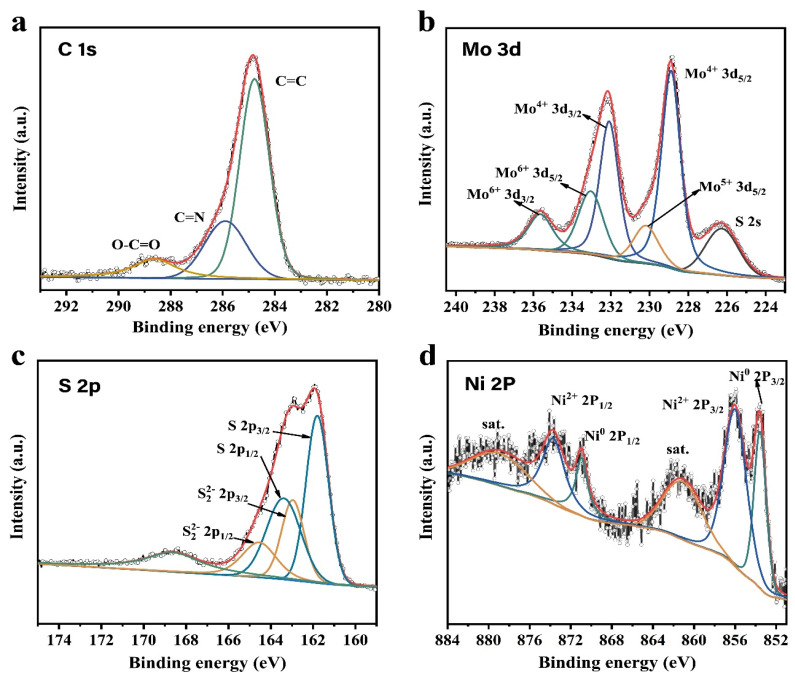
XPS spectra of (**a**) C 1s, (**b**) Mo 3d, (**c**) S 2p, and (**d**) Ni 2p in MoS_2_/rGO/NiS-5 nanocomposite.

**Figure 5 nanomaterials-11-00662-f005:**
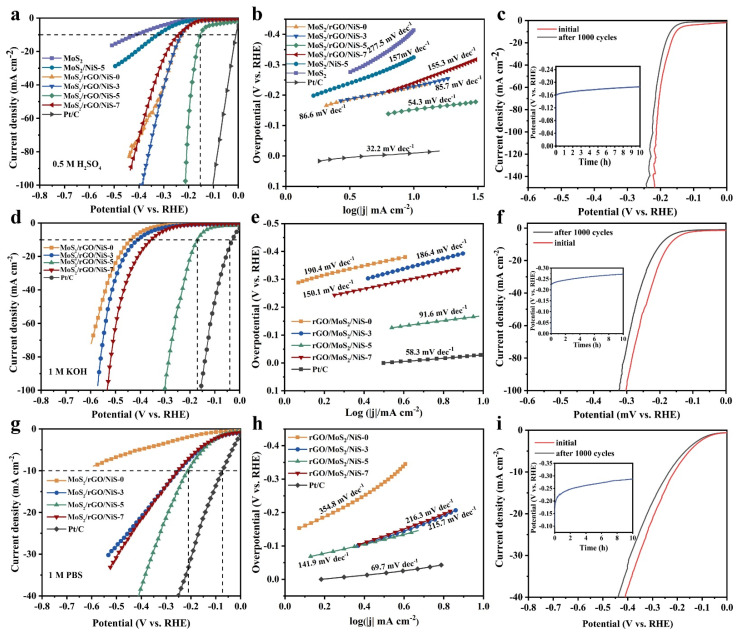
HER catalytic activities of the as-prepared samples. (**a**,**d**,**g**) Polarization curves in 0.5 M H_2_SO_4_, 1.0 M KOH and 1.0 M PBS, respectively. (**b**,**e**,**h**) Tafel plots under 0.5 M H_2_SO_4_, 1.0 M KOH and 1.0 M PBS, respectively. (**c**,**f**,**i**) The polarization curves of MoS_2_/rGO/NiS-5 before and after 1000 cycles of voltammetry in 0.5 M H_2_SO_4_, 1.0 M KOH and 1.0 M PBS were compared, respectively. Their insets were chronopotentiometric curves under 0.5 M H_2_SO_4_, 1.0 M KOH and 1.0 M PBS, respectively.

**Figure 6 nanomaterials-11-00662-f006:**
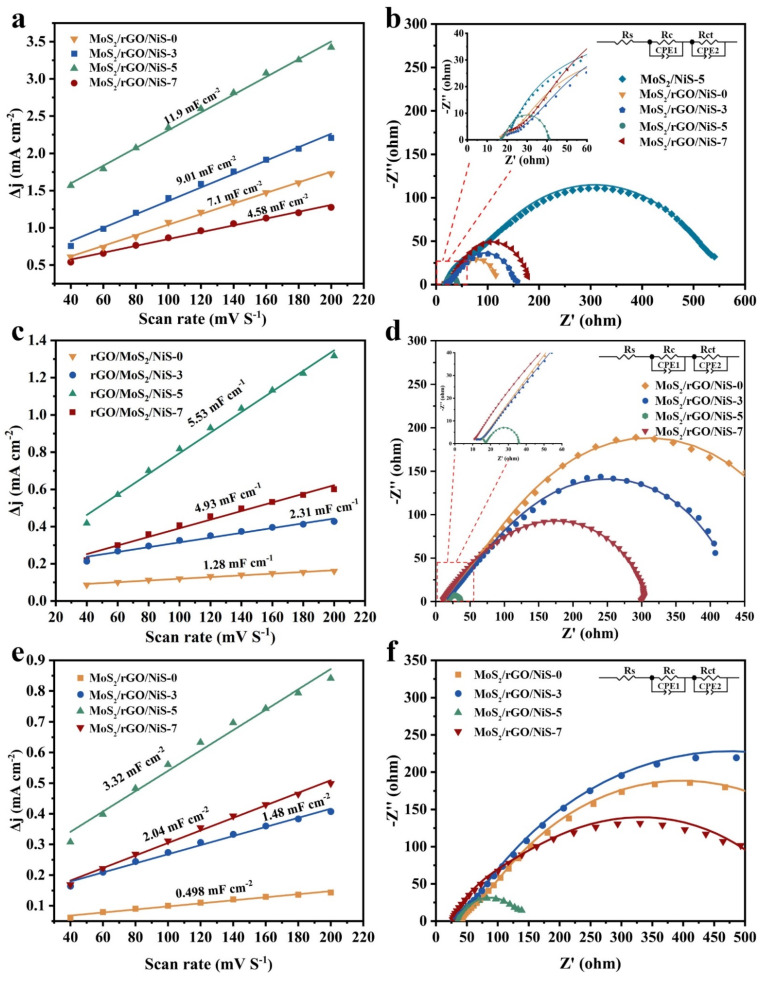
Calculated double-layer capacitance for the samples in (**a**) 0.5 M H_2_SO_4_, (**c**) 1.0 M KOH and (**e**) 1.0 M PBS. The Nyquist plots (solid lines), fitting data (dot) and equivalent circuit (inset) of the samples in (**b**) 0.5 M H_2_SO_4_, (**d**) 1.0 M KOH and (**f**) 1.0 M PBS.

**Table 1 nanomaterials-11-00662-t001:** Comparison of the MoS_2_-based catalyst in acidic and alkaline electrolytes.

Catalysts	Medium	η(mV) ^a^	Tafel Slope (mV/dec)	Loading (mg/cm^−2^)	Substrate	Ref.
MoS_2_/rGO/NiS	0.5 M H_2_SO_4_	152	54.3	0.353	GC	this work
Ag_2_S/MoS_2_/rGO	0.5 M H_2_SO_4_	190	56.0	0.250	GC	[44]
MoN-MoS_2_	0.5 M H_2_SO_4_	190	59	0.250	GC	[46]
CoS_2_-C@MoS_2_	0.5 M H_2_SO_4_	173	61	0.250	GC	[47]
Ag_2_S/MoS_2_	0.5 M H_2_SO_4_	220	42	0.570	GC	[25]
N-C/MoS_2_	0.5 M H_2_SO_4_	185	57	0.105	GC	[38]
MoS_2_/rGO/NiS	1 M KOH	169	91.6	0.353	GC	this work
MoSx@NiO	1 M KOH	406	43	0.706	GC	[48]
MoS_2_@FePS_3_	1 M KOH	175	127	2.000	NF ^b^	[49]
Co_9_S_8_@MoS_2_	1 M KOH	177	83.6	^———^	GC	[50]
Co_3_S_4_/MoS_2_/Ni_2_P	1 M KOH	178	98	0.510	GC	[51]
MoS_2_/rGO	1 M KOH	168	83.9	0.566	GC	[52]

^a^ Overpotential at a current density of 10 mA/cm^2^. ^b^ Nickel foam.

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
