# Peer review of "Defect-Rich Heterogeneous MoS2/rGO/NiS Nanocomposite for Efficient pH-Universal Hydrogen Evolution"

_nanomaterials, 2021, doi:10.3390/nano11030662_

Round 1
Reviewer 1 Report
In this paper, the authors reported the preparation of defect-rich heterogeneous ternary nanocomposites composed of MoS2, NiS, and reduced graphene oxide (rGO) as electrocatalytic catalyst for hydrogen evolution reaction.
The characterization of nanocomposites is performed by various experimental methods such as XRD, Raman spectrum, SEM, HRTEM, and XPS. They investigated the hydrogen evolution reaction activity of the nanocomposites using various electrochemical measurements, and found that the nanocomposites especially MoS2/rGO/Ni-5 shows excellent catalytic performance and stability over a wide range of pH.
Although the structure of nanocomposites is complicated, I think that the reported results will give a good strategy to develop the electrocatalyst for hydrogen evolution reaction.
There are some minor comments as follows.
Minor comments
- Line 86 MoS2/rGO/Ni-X (X= 0,3,5,7)
What is the X ? The meaning of X should be described in the text.
- Line 125
800-100 → 800-1000 ?
- Fig. 4 (d)
Ni 3d → Ni 2p ?
- Line 196
..and 1.0 M KOH,…→… ,1.0 M KOH and 1.0 M PBS,…
- Line 248
The table S4 should be sited in the text.
- Line 272
Rd → Rc ?
- It should be better that some explanation would be added to the following terms for the readers who are not so familiar with electrochemistry.
Overpotential and catalytic activity
Tafel slope and rate limiting steps (Volmer, Heyrosky, Tafel)
Reviewer 2 Report
In this paper, the authors proposed a new defect-rich heterogeneous ternary nanocomposite consisted of MoS2, NiS and reduced graphene oxide (rGO) synthesized using ultrathin αNi(OH)2 nanowires as the nickel source.
In my opinion, the paper can’t be published on the journal in this version, since first of all the manuscript doesn’t respect the minimum requirements of a scientific paper: the section “material and methods” is completely absent and, more important, the authors mentioned in the text supporting information that the reviewer was not able to find as an attachment to the submitted manuscript. Other issues are the following:
- The authors used multiple references in the introduction section: they should try to evidence the contribution to the discussion of the single reference;
- The authors should check the numbering of the sections: in my opinion, if they use the numbered sections, the Introduction should have number 1, results and discussion number 2, and so on;
- The authors should provide more details regarding the synthesis of the catalysts: they mentioned in the text supporting information that the reviewer was not able to find as an attachment to the submitted manuscript;
- The authors should compare the performance of their catalysts with some other ones present in literature and they should add these data in the paper and not in supporting information (the reviewer was not able to see them)
- Which is the duration of each experimental test? Did the authors repeat the experimental tests by using the same catalysts? Did the authors perform any characterization on the spent catalysts?
Round 2
Reviewer 2 Report
The authors well improved the manuscript, which in my opinion can be published on the journal.